# Maternal death surveillance and response system reports from 32 low-middle income countries, 2011–2020: What can we learn from the reports?

Lillian Whiting-Collins[1], Florina Serbanescu[1]*, Ann-Beth Moller[2], Susanna Binzen[1], Jean-Pierre Monet[3], Jenny A. Cresswell[2], Michel Brun[3], on behalf of the World Health Organization's MDSR Reporting and Monitoring Working Group¶

1 Division of Reproductive Health, Centers for Disease Control and Prevention, Atlanta Georgia, United States of America, 2 Department of Sexual and Reproductive Health and Research, World Health Organization, Geneva, Switzerland, 3 Technical Division, United Nations Population Fund, New York, New York, United States of America

☽ These authors contributed equally to this work.
¶ Membership of the World Health Organization's Maternal Death Surveillance and Response Technical Working Group for Reporting and Monitoring is provided in the Acknowledgments
* fxs7@cdc.gov

**Data Availability Statement:** The data is available at: https://ghdx.healthdata.org/series/maternal-death-surveillance-efforts-sources.

## Abstract

Maternal Death Surveillance and Response (MDSR) systems generate information that may aid efforts to end preventable maternal deaths. Many countries report MDSR data, but comparability over time and across settings has not been studied. We reviewed MDSR reports from low-and-middle income countries (LMICs) to examine core content and identify how surveillance data and data dissemination could be improved to guide recommendations and actions. We conducted deductive content analysis of 56 MDSR reports from 32 LMICs. A codebook was developed assessing how reports captured: 1) MDSR system implementation, 2) monitoring of maternal death notifications and reviews, and 3) response formulation and implementation. Reports published before 2014 focused on maternal death reviews only. In September 2013, the World Health Organization and partners published the global MDSR guidance, which advised that country reports should also include identification, notification and response activities. Of the 56 reports, 33 (59%) described their data as incomplete, meaning that not all maternal deaths were captured. While 45 (80%) reports presented the total number of maternal deaths that had been notified (officially reported), only 16 (29%) calculated notification rates. Deaths were reported at both community and facility levels in 31 (55%) reports, but 25 (45%) reported facility deaths only. The number of maternal deaths reviewed was reported in 33 (59%) reports, and 17 (30%) calculated review completion rates. While 48 (86%) reports provided recommendations for improving MDSR, evidence of actions based on prior recommendations was absent from 40 (71%) of subsequent reports. MDSR reports currently vary in content and in how response efforts are documented. Comprehensive reports could improve accountability and effectiveness of the system by providing feedback to MDSR stakeholders and information for action. A standard

**Funding:** The authors received no specific funding for this work.

**Competing interests:** The authors have declared that no competing interests exist.

reporting template may improve the quality and comparability of MDSR data and their use for preventing future maternal deaths.

## Introduction

The global maternal mortality ratio (MMRs) has declined over the last two decades, from 342 (UI 80%: 327–368) maternal deaths per 100,000 live births in 2000 to 211 (UI 80%: 199–243) deaths per 100,000 live births in 2017 [1]. However, this decline is uneven. Striking regional disparities remain, with MMRs exceeding 500 deaths per 100,000 live births in several Central and Western African countries [1]. Globally, progress is not at the pace needed to achieve the Sustainable Development Goals global target of 70 maternal deaths per 100,000 live births by 2030 [2]. In striving to meet this goal, the Maternal Death Surveillance and Response (MDSR) approach is an important strategy which may aid efforts to reduce maternal mortality. The purpose of the MDSR is to end preventable maternal mortality through the intentional use of data to guide policies and public health programs. MDSR provides the feedback loop that may connect information on maternal deaths with action to prevent future maternal deaths through improved access to, utilization of and provision of timely and quality maternal health care. This evidence-to-action cycle may help reducing the local and global burden of maternal mortality [3].

In recent years, use of MDSR has expanded across low-and middle-income countries (LMICs), where 99% of annual global maternal deaths occur [1,4]. In September 2013, the World Health Organization (WHO) and partners published the technical guidance that described MDSR system concepts and provided instructions for establishing MDSR at country level [3]. This guidance outlines the essential MDSR components as follows:

i. Ongoing *identification* and *notification* of maternal deaths

ii. *Review* of maternal deaths by local committees

iii. *Analysis* of aggregated data from reviews

iv. *Interpretation* of findings and *development of recommendations*

v. *Implementation* and *monitoring* of recommendations.

WHO guidance emphasizes the importance of making maternal death a notifiable event (reported to the health information system) in all countries, ensuring timely notification (within 24 to 48 hours) and conducting a thorough investigation and analysis of all maternal deaths, followed by recommendations for clinical and community actions. It also stresses that identification of all maternal deaths requires information from both health facility and community sources. Equally important is the need to identify all deaths of women of reproductive age, assess if they have occurred during pregnancy or within 42 days of the end of pregnancy (suspected maternal death), and notify all suspected maternal deaths to reduce under-reporting and misclassification. Notification benefits from "zero reporting," an active process of routinely notifying the number of suspected maternal deaths, whether any occurred or not. The guidance delineates processes for maternal death review committees, and suggests model data analyses and presentation of results, as well as possible recommendations based on results. Monitoring and evaluation of the MDSR system is a key component to improve the quality and completeness of information. Reports are generated by in-country teams of health

scientists and physicians, typically convened by the Ministry of Health to conduct a thorough review of the MDSR system and to provide recommendations for system improvement.

The guidance also provides an outline of sections to include in an annual MDSR report, noting that reporting may occur in the facility, at local (e.g., district) and/or national levels, and makes recommendations for disseminating MDSR findings. The guidelines specifies that MDSR reports have to be shared with the facilities or communities where information was gathered, and that confidentiality of the information must be preserved to prevent the use of findings in litigation against facility staff. Those with the position and authority to take action such as local and national governments, public health departments and religious leaders, should also receive the reports [3].

In 2015, WHO established an MDSR Global Technical Coordination Group charged with providing guidance to countries on MDSR implementation, analyses, reporting, as well as monitoring and evaluation. Its sub-working group on "Reporting and Monitoring" provides countries with support and guidance for using MDSR data that may in turn help reduce preventable maternal deaths. Since its inception, this group has tracked MDSR reporting efforts worldwide. However, there remains a gap in understanding of how country MDSR reports can be best utilized among key partners to inform response efforts and improve maternal health outcomes.

Our objective was to review existing MDSR country reports issued in LMICs from 2011 to 2020 and document: 1) MDSR implementation; 2) type of recommendations and implementation of responses; and 3) the extent to which countries monitor their MDSR systems. By conducting this review, we aim to detect core elements and indicators that could be included in a reporting template to facilitate evidence-based reporting, and also to identify how surveillance data collection, review, analysis, dissemination and response could be improved at country and global levels based on WHO's guidance.

## Methods

MDSR reports were gathered from countries by the United Nations Population Fund (UNFPA), WHO, and MDSR Reporting and Monitoring working group members through purposive sampling. MDSR teams in LMICs with known functioning MDSR systems were contacted and invited to share their reports for this study. Reports were included if they met the following criteria: contained data on maternal death reporting or review at any level of the health system; were published between 2011 and 2020; and were in English or French. Reports that investigated maternal health care without providing data on reporting or review of maternal deaths were excluded. Most reports (68%, n = 38) that met the inclusion criteria were provided by 24 out of 27 countries who responded affirmatively to UNFPA's Maternal Health Thematic Fund survey's question inquiring if the countries issued MDSR reports [5]. The MHTF survey, conducted annually in 39 countries since 2015, is the only direct source of information about the practice of issuing periodic MDSR reports in LMICs. According to the MHTF, the 27 countries with periodic reports have responded that they issued 53 reports in 2014–2020, including the 38 reports that we were able to obtain (response rate = 38/53 = 72% of MHTF reports). It remains unclear if the MHTF reports that we were unable to obtain (n = 15) would have met the inclusion criteria of our study. The remaining 18 reports (32%) included in our analysis were identified by WHO and its MDSR working group from LMICs that do not participate in the MHTF surveys.

We conducted a qualitative deductive content analysis to identify themes and patterns of core elements in the reports. Researchers from the MDSR Reporting and Monitoring working group developed a codebook of 47 items (codes) grouped by the following topics: MDSR data

collection methods, self-assessed completeness and self-assessed quality; MDSR data analysis (sociodemographic characteristics, pregnancy outcomes, underlying cause of death and contributing factors); description of national MDSR policies, system structure and monitoring; formulated recommendations (reviewed using SMART criteria of specific, measurable, attainable, relevant and timebound); and report structure and content. We assessed structure and content using the components outlined in the WHO guidance on dissemination and reporting [3]. Five researchers used a shared Microsoft Excel document to extract data on 47 items from each report using a data extraction guide. (S1 Text) Each report underwent independent review by two researchers to verify consistency and a third researcher weighed in to resolve disagreements. For the 12 countries with multiple reports in the study period, sub-analysis was done using Atlas.ti qualitative software (v9) to investigate how subsequent reports built upon prior recommendations.

We organized our findings along three objectives: 1) description of MDSR implementation, including details on implementing each element of the MDSR cycle [3]; a qualitative analysis of recommendations and responses and how they were implemented in a sub-set of countries that issued follow-up reports; and a summary of the efforts to monitor the surveillance systems, as captured in the MDSR reports.

## Results

### Overview of reports analyzed

In total, 56 reports met the inclusion criteria, representing 32 countries that published reports on maternal death during the study period, 12 of which published multiple reports in this period (Table 1). One-third (32%, n = 18) also included data on perinatal deaths.

Prior to 2014, reports presented information on conducting Maternal Death Reviews (MDR) and Maternal and Perinatal Death Reviews (MPDR), which focused on examining causes of and circumstances surrounding maternal/perinatal deaths that occurred primarily in health facilities to identify and address gaps in quality of care. They did not report on surveillance activities such as identification and notification of all maternal deaths, or on response activities such as the generative steps taken to develop recommendations based on the analysis of the reviews and how these recommendations were implemented and monitored. Most reports published in 2014 or after focused on MDSR and Maternal and Perinatal Death Surveillance and Response (MPDSR), as more countries transitioned to surveillance systems that stress the need to collect data on all maternal deaths using clearly defined data sources and processes for identification and notification and to respond to each maternal death with appropriate actions. However, some countries continued to publish reports about MDR only, lacking details on any implementation of surveillance activities. By 2017, most countries had expanded the scope of their reports to include surveillance data. Trends in types of reports published over time are shown in Fig 1.

Depending on the type of the report, various sources of data on maternal deaths were mentioned: maternal death notifications, MDRs (with and without verbal autopsy, which is a structured interview of the family or community members who were present at the time of death), death certificate reviews, facility-based surveys and reports, patient records, national health information system data, weekly integrated disease surveillance and response data and semi-structured interviews with stakeholders.

Of the 56 reports, 9 (16%) described their data as "complete" and 14 (25%) reports did not discuss data completeness. Methods of assessing data completeness varied and were not fully explained. More than half of the reports 33 (59%) highlighted underreporting of maternal deaths. Eight reports (14%) indicated that their data quality was affected by inconsistencies and/or missing variables.

**Table 1. Maternal death data reports by country and publication year.**

| Country | 2011 | 2012 | 2013 | 2014 | 2015 | 2016 | 2017 | 2018 | 2019 | 2020 |
|---|---|---|---|---|---|---|---|---|---|---|
| Bangladesh | | | | | | | | | | ▓ |
| Benin | | | | | | | | ▓ | | |
| Burkina Faso | | | | ▓ | | | ▓ | | | |
| Congo | | | | ▓ | | | | | | |
| Côte d'Ivoire | | | | | ▓ | | 2 | | | |
| Democratic Republic of Congo | | | | | | | ▓ | | | |
| Ethiopia | | | | | | ▓ | ▓ | | | |
| Ghana | | | | | | ▓ | | | | |
| Jordan | | | | | | | | ▓ | | |
| Kenya | | | | | | | * 2 | | ▓ | |
| Lao PDR | | | | ▓ | | | | | | |
| Lesotho | | | | ▓ | | | | | | |
| Liberia | | | | | | ▓ | | | | |
| Madagascar | | | | | | | | | ▓ | |
| Malawi | | | ▓ | | | | * 3 | | | |
| Mauritania | | | | | | | | | ▓ | |
| Morocco | | | ▓ | | ▓ | | | | | |
| Myanmar | | | ▓ | | | | | | | |
| Namibia | ▓ | | | ▓ | | ▓ | | | | |
| Nepal | | | | ▓ | | | | | ▓ | ▓ |
| Niger | | | | | ▓ | | | | | |
| Nigeria | | | * | | | | | | | |
| Rwanda | | | | | ▓ | | | | | |
| Senegal | | | | | ▓ | | | | | |
| Sierra Leone | | | | | | | 2 | ▓ | ▓ | |
| Sri Lanka | | | | ▓ | | | | | | |
| Sudan | | | | | | ▓ | | | | |
| Eswatini | | | | ▓ | ▓ | ▓ | | | | |
| Timor-Leste | | | | | | | ▓ | ▓ | ▓ | |
| Togo | | | | | | | | ▓ | | |
| Uganda | ▓ | | ▓ | | | | ▓ | | 2 | |
| Zambia | | | | | | | | ▓ | | |

*Indicates a subnational report. Numerals in cells indicate more than one report in that year (a national report and 1–2 subnational reports or 2 national reports).

Thick line between 2013 and 2014 signifies before and after the publication of WHO's guidance on MDSR.

Note: A report's content may span one or more years.

Most reports (66%, n = 37) described their national MDSR policies. Over half (54%, n = 20) that included a description of policies, did not mention whether reporting was mandatory, 46% (n = 17) specified mandatory reporting, including 2 reports enforcing zero reporting (in which systems actively report "0" when no women had died during, pregnancy, childbirth or postpartum). Among reports that detailed MDSR policies, 41% (n = 15) did not mention timeliness of notification. When reports did provide a timeframe in which maternal deaths must be notified, most mandated a 24–48 notification timeframe.

**Maternal death surveillance and response implementation.** Maternal deaths that occurred in health facilities and in the community were included in 31 reports (55%), while 25 (45%) presented data on facility deaths only (Fig 2). Most reports (80%, n = 45) included the

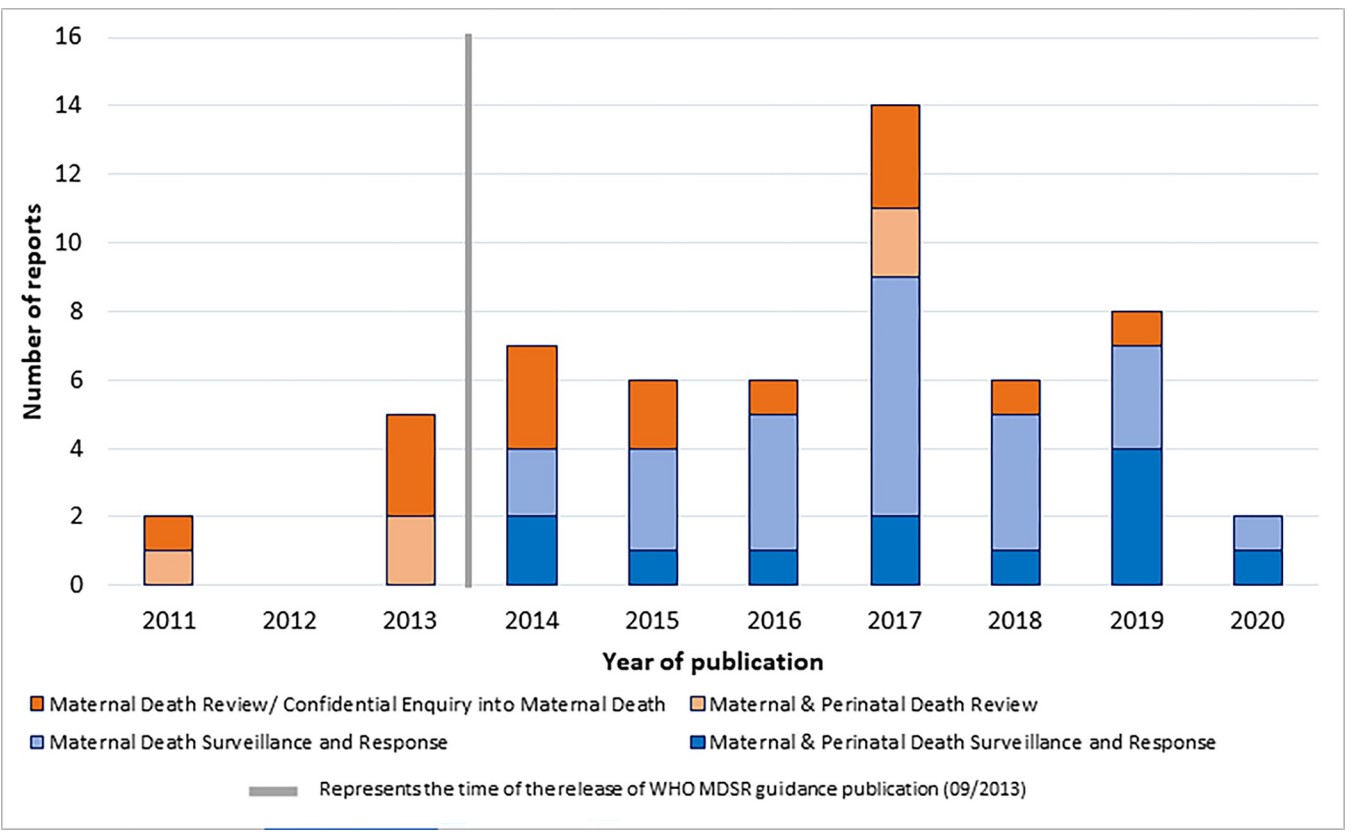

**Fig 1. Reports on maternal death data submitted by 32 low- and middle-income countries by type of report and publication year, 2011–2020 (N = 56).**

total number of maternal deaths that were notified during the reporting period. Over half of reports (59%, n = 33) presented data on maternal death reviews. Almost half of reports (48%, n = 27) provided information on both national and subnational MDSR committees, including 27% (n = 15) of reports listing the professionals involved in these committees and 25% (n = 14) identifying a focal point or MDSR leader. Other reports did not mention the committees at all (20%, n = 11), a national committee was mentioned in 11 reports (20%) and a subnational committee was mentioned in seven (13%).

Most reports (73%, n = 41) did not provide details on the processes carried out by maternal death review committees. Among reports that summarized the review process (27%, n = 15), level of detail and completeness of description varied. One report highlighted limitations in the review process due to poor documentation about the death and lack of guidance for facility review committees; another provided details on the data sources and cross-checking prior to review; and a third detailed timeframe and procedures for reporting the committee findings to health officials.

**Response formulation and implementation.** The type of responses recommended in the reports reflected countries' efforts to develop remedies relevant to their situations to address the issues they identified at every organizational level. In 80% of reports (n = 45), factors contributing to maternal deaths were identified at community, facility, local, regional and national levels, with recommendations often addressing each underlying issue (Fig 3). Nearly all reports (86%, n = 48) included recommendations for improving the MDSR system in the country of interest. These recommendations, however, often lacked the "SMART" criteria of having

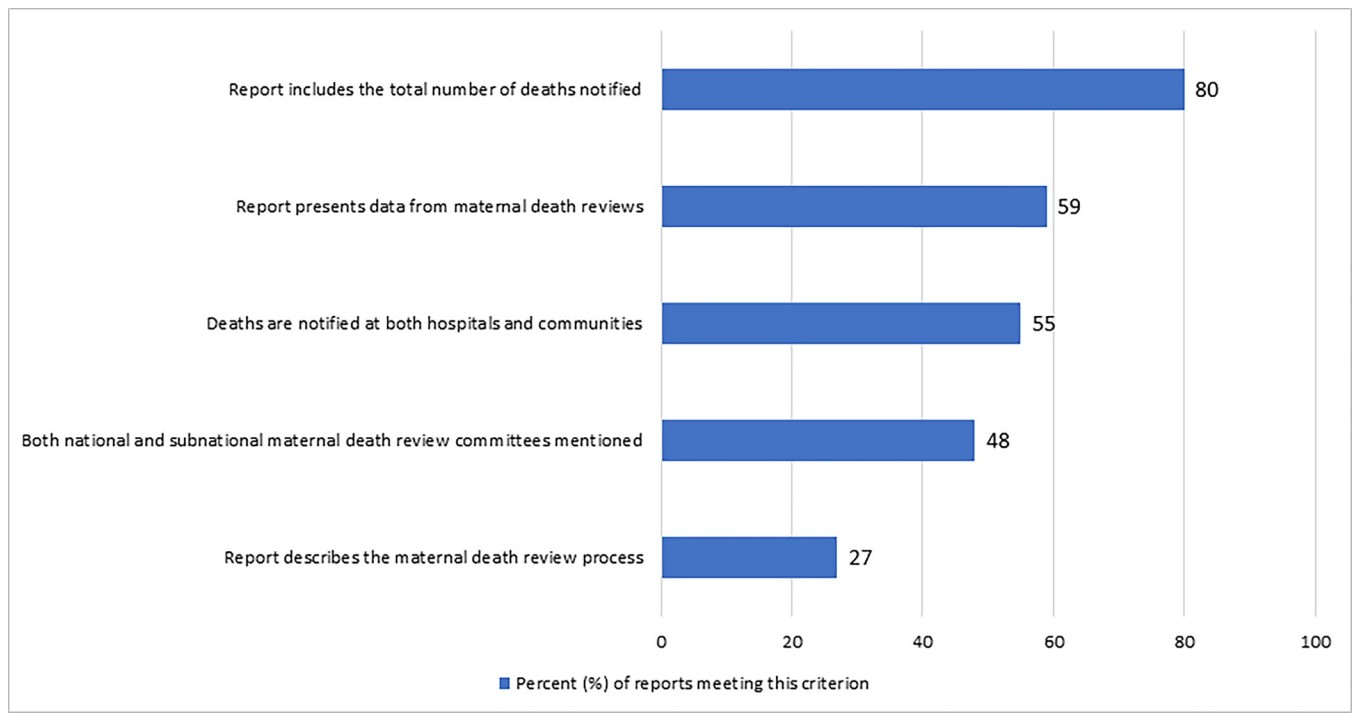

**Fig 2. Percentage of maternal death data reports (2011–2020) that include selected aspects of the Maternal Death Surveillance and Response implementation (N = 56).**

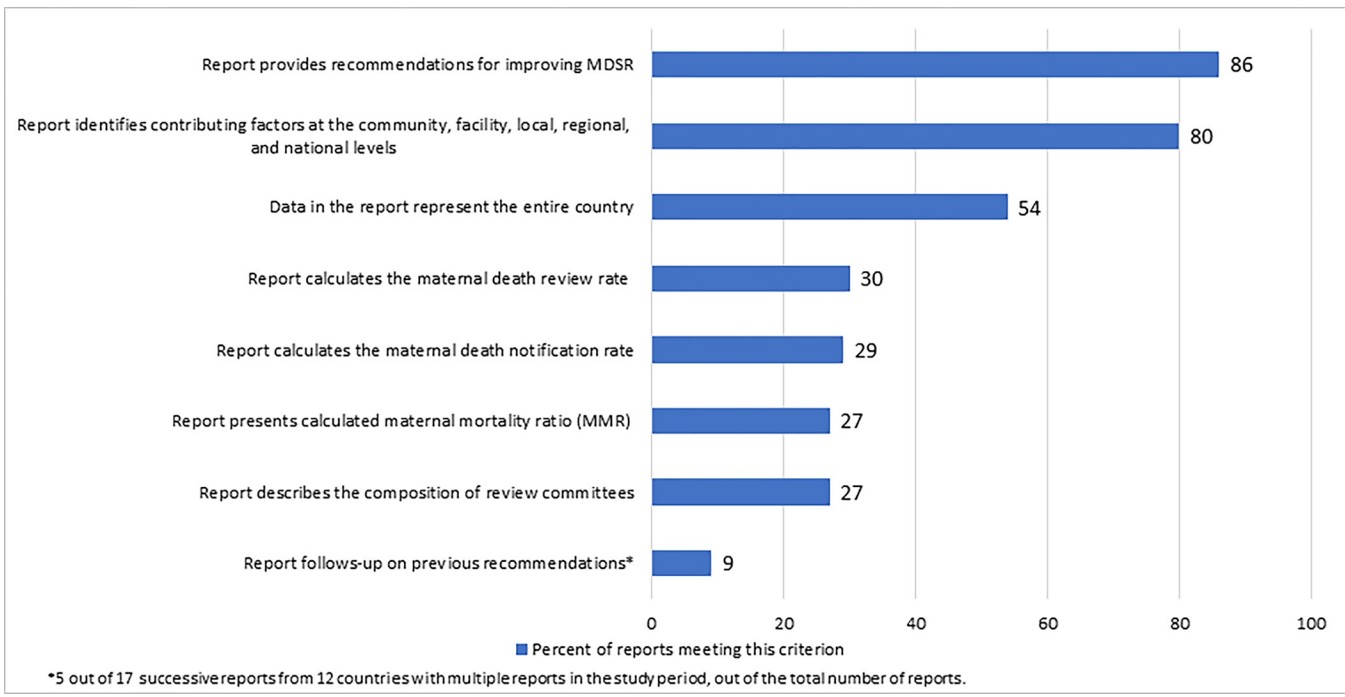

**Fig 3. Percentage of maternal death data reports (2011–2020) that include selected criteria outlined in the Maternal Death Surveillance and Response guidance [3] (N = 56).**

*s*pecific, *m*easurable, *a*ttainable, *r*elevant, and *t*imebound objectives, with most recommendations failing to be timebound or measurable. About a third (39%, n = 22) of reports included recommendations that were specific. Table 2 provides examples from the country reports of recommendations relating to improve both quality of maternity care and the MDSR system itself, organized by community, facility, district/regional/zonal and national levels.

**Table 2. Examples of issues and recommendations identified in maternal death data reports, 2011–2021.**

| Issues identified | Example recommendation |
|---|---|
| *Community level* | |
| Lack of community engagement in MDSR. | "Community participation and involvement needs to be ensured by engaging Paramount Chiefs, district councils and other community leaders to integrate MDSR into the existing community structure. These leaders can encourage facility births and improved reporting of suspected maternal deaths from their communities and discontinue punitive bylaws that hinder community participation in MDSR." [6] |
| No data on community notifications on maternal and perinatal death. | "Orient the Village Health Teams on reporting tools and MPDSR guidelines." [7] |
| Low utilization of maternal health care. | "Communities should be educated about early ANC attendance. Health care providers should ensure that antenatal care is accessible and welcoming so that all women can reach them easily and early in their pregnancy." A significant number of women present for confirmation of pregnancy early and measures should be put in place to link pregnancy confirmation visits with comprehensive ANC content (e.g., nutritional interventions, screening, diagnosis and disease prevention). "It is also important that women are seen at least once by a medical officer/nurse clinician/advanced midwife during the course of pregnancy. Health care providers should ensure that no woman who has attended ANC gets into labor without any ANC investigations being done and acted on for those that are suggestive of complications." [8] |
| Disparities in maternal deaths and reporting among communities. | "DHMTs should outreach to chiefdoms with large numbers of maternal deaths to better understand reasons for death and develop strategies to prevent more deaths. DHMTs should also outreach to community leaders in silent chiefdoms to ensure that maternal deaths are not happening un-reported in these chiefdoms." [9] |
| *Facility level* | |
| Health worker shortage and poor motivation. | "Increase number of skilled birth attendants (staff motivation, address motivational factors at all levels). Effect result-based appraisal systems, e.g., promotions based on merit to improve staff motivation." [10] |
| Challenges in managing obstetric complications (i.e., prompt reaction, life-saving skills, lack of critical thinking) which affects proper and timely decision making. | "Improve staff performance management systems including accountability mechanisms." [11] |
| Misclassification of maternal deaths in health records. | "Clinicians from all hospitals who care for pregnant women should be trained in ICD-10 classification of maternal deaths to allow for accurate classification of deaths in the MDSR system." [6] |
| Insufficient data due to lack of MDSR data from private and missionary hospitals. | "The MDSR program should be rolled-out to private and missionary hospitals with provision of adequate support from the national and district levels (forms, training, meetings etc.)." [9] |

*(Continued)*

**Table 2.** (Continued)

| Issues identified | Example recommendation |
|---|---|
| *District/regional/zonal level* | |
| Lack of access and availability of blood products | "Improve access and availability of blood products, supplies and consumables through equipping training and onsite supervision of all CEmONC facilities in collaboration with Regional Blood Banks." [7]<br>"Health facilities should have reliable access to blood/blood products 24 hours a day." [12]<br>"The national safe blood system should be strengthened so that safe blood is available to treat postpartum hemorrhage." [9] |
| Discrepancies in data between DHIS2* and maternal death notifications. | "Ensure monthly consultation between the MDSR focal point and the health zone statistician for joint validation of maternal death notifications and the DHIS2, to avoid differences in data." [13] |
| The technical capabilities of providers on MDSR committees are insufficient to make quality reviews. As a result, the recommendations do not address the problem, which negatively affects response efforts. | "Establish a clinical mentor in each health zone to support death reviews and response organization." [14] |
| No data on the percentage of districts with functional maternal/perinatal death surveillance and response committees. | "Establish where committees don't exist, reorient where they do exist and distribute committee guidelines. Conduct a rapid annual assessment to establish the existence and functionality of district MPDSR committees. Coach and support supervisors from regional MPDSR committees." [7] |
| *National level* | |
| A need to strengthen the "R" in MDSR. | "At national level there should be an annual symposium at which the latest national data on maternal death are. presented with appropriate analysis. At the meeting there will be capacity building of participants including improving the standards of death reviews and improving the standards of responses." [15] |
| A need for a national MPDSR review committee. | "Elaborate terms of reference and work plan to render the national MPDSR review committee functional." [16] |
| A need for training materials to improve emergency obstetric and intrapartum care. | "The Ministry of Health and the Society of Obstetricians and Gynecologists should develop user-friendly guidelines, laminated flow chart and pocketbooks for EmONC and intrapartum care in relevant languages to be distributed to provinces and subsequently to districts and health centers." [17] |

Acronyms: ANC, antenatal care; CEmONC, comprehensive emergency obstetric and newborn care; DHIS2, district health information system version 2; DHMTs, district health management teams; EmONC, emergency obstetric and newborn care; MDSR, maternal death surveillance and response; MPDSR, maternal and perinatal death surveillance and response.

Implementation of recommended responses was difficult to assess. Twelve countries published multiple reports in the study period; however, tracking of prior response efforts and progress over time was present in follow-up reports from three of these 12 countries only. In one country, follow up reports described how the MDSR system has evolved and responded to learning over time, yet it stated that it cannot verify if districts implemented recommendations from the MDSR reviews. Notably, this country included action items with updates on progress made from one report to another and the status of implementing each item. A second country included a table of indicators to track progress towards meeting National MDSR Guideline standards from one report to another. Allocation of timebound responsibilities to a particular person or group accompanied each recommendation in all three countries.

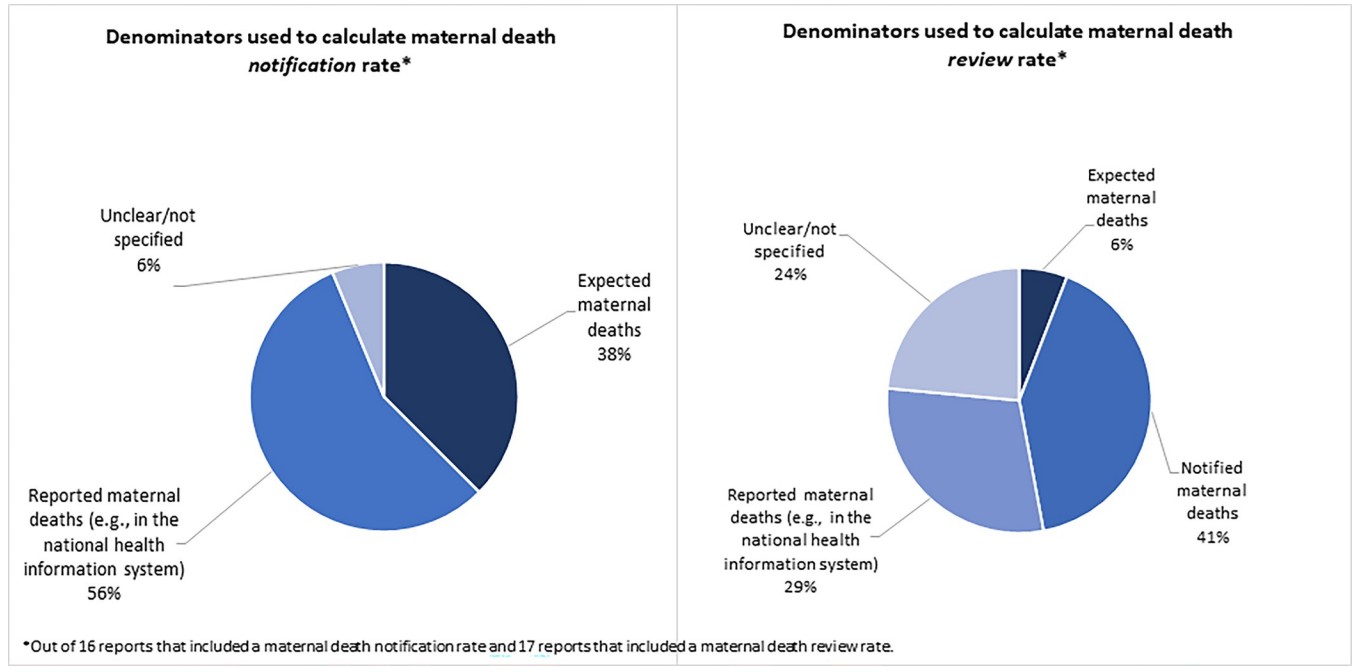

**Fig 4. Distribution of maternal death data reports (2011–2020) presenting maternal death notification rate (N = 16) and maternal death review rate (N = 17) by denominator used for the rate.**

**Monitoring and evaluation of the MDSR system.** Over half of reports (54%, n = 30) presented maternal death data from the MDSR system for the entire country, but data were only presented for a subset of reporting regions in 44% (n = 26) of reports (Fig 3).

Indicators for maternal death notification were defined in various ways, although notification data were missing from several reports (20%, n = 11). For reports that included a notification rate (29%, n = 16), meaning the proportion of maternal deaths notified out of the total known or estimated from other sources, the denominator for this calculation came from a range of sources (Fig 4). Just over half (52%, n = 29) provided the number of maternal deaths notified in the reporting period without calculating a notification rate; instead, notification numbers were displayed graphically to demonstrate changes in the count of deaths notified over time. MDSR data were used to calculate a maternal mortality ratio using the number of notified deaths and the number of expected live births, in 27% (n = 15) of reports (Fig 3).

Information on maternal death review was missing from almost half (41%, n = 23). In reports that included information on reviews (59%, n = 33), only about half (30%, n = 17) calculated a maternal death review rate (the proportion of maternal deaths reviewed out of the total maternal deaths known or estimated from other sources) (Fig 3). The type of denominator used in the review rate calculation varied across reports, as shown in Fig 4 (Fig 4).

Few reports provided information on additional measures of tracking MDSR activities, including those for quality of maternal death reviews, although lack of data on review processes was often cited as a challenge.

Analyses specifying deaths by maternal characteristics and care received were presented in over half of reports (63%, n = 35), with sub-analyses related to factors shown in Box 1. Less than half (41%, n = 23) of reports presented deaths by pregnancy outcome such as live birth, stillbirth, spontaneous abortion, or induced abortion. Most reports (89%, n = 49) presented both direct and indirect obstetric causes of maternal death. The remaining reports either

provided data on direct causes only (7%, n = 4) or the causes of death were not mentioned (4%, n = 2). A handful of reports assessed trends in maternal deaths over time by cause of death and by characteristics. More commonly, reports discussed medical and non-medical factors that contributed to maternal deaths (80%, n = 45) (Fig 3), with 40% (n = 23) of reports presenting contributing factors according to the Three Delays model (deaths associated with the delay in the decision to seek care, delay in reaching care, and delay in receiving adequate care) [18]. Reports lacked descriptions of changes in clinical practice or in the surveillance system over time in response to recommendations from the review committees.

---

**Box 1. Most common demographic and access to care characteristics used for analysis in maternal death data reports (2011–2020) (N = 56)**

Deaths were assessed by the following

*Maternal characteristics*

- Age

- Parity

- Occupation

- Socioeconomic status

- Education level

- Ethnicity/caste

- HIV status

- Presence of preexisting conditions

*Care received by the women*

- Antenatal care: any ANC, number of visits.

- Condition on admission to the hospital (e.g., in labor or not; with signs and symptoms of obstetric complications or not). Some reports include "dead on arrival" or "unconscious at arrival."

- Direct and indirect complications during pregnancy and childbirth (obstetric hemorrhage, pregnancy induced hypertension, obstructed labor, uterine rupture, abortion, puerperal sepsis, other direct complications, indirect complications)

- Mode of birth

- Type of birth attendant

---

Because it was not addressed in any report, it is impossible to know whether or how the recommendations of the review committees were enacted. However, one report details an evaluation of the country's MPDSR system, which highlights the challenge of tracking the implementation of maternal death review committee recommendations when there are no mechanisms in place to track status of response actions or the assignments to focal points

responsible to complete the actions. A government health worker interviewed for the report stated: "There is no effective feedback mechanism from higher-level regarding action plan implementation. Also, there is no regular meeting with the MPDSR technical working group to track the action plans." A doctor added: "Follow-up on action plans is the most important part of MPDSR system but sadly, this activity does not happen so often. Hence, the most important part is missed." [12].

## Discussion

Our findings provide insight into the status of MDSR reporting in LMICs between 2011–2020, highlighting areas of progress as well as opportunities for major improvements. We found that starting with 2014, more countries started to report on surveillance and response activities in addition to maternal death reviews, consistent with trends observed in the literature [3,4]. The reports included in this assessment represent a range in terms of stages of MDSR implementation, with several reports detailing MDSR in only one region, while others present data from the entire country; just over half included reports from both facilities and community sources. Despite having national policies and a national system in place, many countries still struggle to enforce policies and implement the system in all regions. Lack of funds, human resources, adequate supervision and delays in reporting are reasons given for not being able to have deaths reported from all regions at the time annual reports are prepared. Reduced coverage of maternal death notification and review could be interpreted as a reflection of the phase of MDSR scale-up rather than poor system performance but could also indicate poor implementation of scaled-up systems. Underperforming systems are likely to have an almost negligible effect on improving maternal health.

Countries reported using a variety of data collection methods, both routine and investigative, that typically culminate in indicators of maternal death notification and maternal death review. In reports that included a description of the MDSR policy environment, details were often lacking; for example, only two reports, both from the same country, indicated that the MDSR system had a policy of mandatory zero reporting. Nearly all reports provided results from sub-analyses of the maternal death causes, maternal characteristics, pregnancy outcomes and care received prior to death; however, the extent to which these data informed response efforts was unclear. Inclusion of recommendations was nearly universal, yet the strength of these recommendations (i.e., were they specific, measurable, attainable, relevant, or time-bound) was limited. Although recommendations can only improve an MDSR system when they are enacted, only 3 out of 12 countries that issued follow-up reports provided information on whether or how the recommendations were carried out or whether they had an effect.

Most reports do not provide insight into using their data for response efforts, and when present the list of next steps is not clearly rooted in the data presented. Without sufficient explanation of how data from the reports are being used, it is difficult to track learning over time or to identify factors that inhibit or support MDSR system scale-up. This observation echoes recent findings by Kinney et al. in their scoping review of MPDSR implementation: that lessons learned from studies on M(P)DSR implementation have changed little over the past decade [3]. Despite increased reporting over time, the lack of evidence that the data are used to inform action may impede MDSR evolution and scale-up.

Some studies have highlighted the inhibitive role of a culture in which frontline health care providers fear repercussions or punitive measures after maternal death occurs, presenting barriers to maternal death notification, review and use of data [19,20]. In Kenya, Smith et al. described health facility staff's hesitancies about releasing case files on women who died, despite a "no name, no blame" approach which anonymized cases to reduce the culture of

blame and shame in facility-based maternal death reviews [21,22]. On the other hand, Bakker et al.'s qualitative study on Malawi health workers' perceptions of maternal death audit demonstrated that maternal death review can be viewed as positive and motivating if the focus is on learning rather than blaming [23]. Shifting the approach to one of learning lessons from the deaths and making improvements informed by those experiences can help to promote use of maternal death surveillance data and information from maternal death reviews in other settings.

Our qualitative analysis of recommendations included in the MDSR reports revealed that some countries are formulating clear, evidence-based recommendations for facility and community actions, rooted in the review of the maternal deaths as part of the surveillance process. It has been argued that LMICs need to bypass the resource-intensive activity of developing a targeted response effort to prevent future maternal deaths based on findings from maternal death reviews, and instead directly implement well-known interventions to reduce maternal deaths [24]. Others have counter-argued that such initiatives should, and often do, occur in tandem with MDSR efforts to improve accountability [3,25]. In countries with a high burden of maternal mortality, reviewing at least a subset of maternal deaths and developing responses based on committee findings can contribute to improve the country's maternity care. MDSR reports, when shared and acted upon, could facilitate strengthening of this feedback loop to ensure sustainability of MDSR efforts. Sierra Leone's 2018 and 2019 reports are a prime example of how such documents can be linked to provide both context and follow-through on progress of implemented response items. Only three countries with multiple reports included information indicating what entities were responsible for enacting the recommendations. These findings highlight areas for improvement and are consistent with identification of gaps in the follow-on activities to maternal death reviews, such as lack of community or civil society involvement in implementing recommendations, as identified by Bandali et al [25]. When maternal death reviews are of high quality, their comprehensive information informs more effective recommendations and stronger response actions and can effectively reduce maternal mortality [26].

Our findings revealed inconsistencies in how key MDSR indicators are tracked, measured and reported. For example, maternal death notification rates were calculated in several reports, but with different denominators, making it challenging to compare across settings. This was also observed for calculation of the maternal death review rate. To facilitate such comparisons, countries can use standardized and well-defined indicators, such as those recommended by WHO and thus may foster examination of trends and global comparisons [4]. Use of coverage indicators, which use global estimates of maternal death to measure coverage rates of notification and review, may improve monitoring by indicators and strengthen the surveillance system overall [27,28]. To reduce the administrative burden of developing lengthy MDSR reports and to facilitate dissemination of findings in an accessible, targeted way, a core reporting template outlining what indicators should be included, how they should be measured and establishing a way to link reports over time to track progress can inform efforts in the global community. Beyond measuring MMRs and trends, national reports offer opportunities to track other aspects of MDSR, including quality of maternal death reviews, quality of actionable responses and interventions taken to strengthen the ability of the MDSR system to improve maternal health. While the literature on MPDSR implementation is growing in LMICs, particularly in Africa, Kinney and colleagues found that most studies focused on "tangible inputs from a care provision lens," did not systematically report on all steps of the surveillance cycle or on the system as a whole and often overlooked the role of implementers and communication channels within MPDSR or between MPDSR and the health system [4]. Strengthening the quality of national reports, intended to report on the entire surveillance cycle, its monitoring

and its iterative improvements, may be able to address some of the implementation gaps identified by our study and in the literature.

MDSR reports across countries have not been previously reviewed using a structured approach. Our synthesis of content and structure of MDSR reports identified positive features and practical considerations that could be scaled up to help improve reporting and facilitate learning lessons both at country and global levels. It also identified gaps and missed opportunities for improving clinical practice and surveillance. The process that we used to gather MDSR reports through the UNFPA country offices and the WHO MDSR Reporting and Monitoring Working Group members' connections to country MDSR systems in LMICs could be considered a strength of this study. However, it is possible that this method of collection may have missed some MDSR reports from countries without UNFPA or WHO connections. Further, our analyses predominantly included reports from sub-Saharan Africa, which prevents us from examining regional differences. Also, there may be additional reports for 2020 that were not captured in our analysis, as publication of MDSR reports may be delayed by one or even two years. Lack of access to follow-up programmatic or policy documents is a limitation of this study, as reports do not include information on what happened *after* the report's dissemination. It is possible that more countries addressed recommendations formulated in the MDSR reports than those identified in our analysis. Without adding a core section that describes actions taken following the previous report makes it difficult to interpret if recommendations were implemented, to what extent and their resulting outcomes. Reviewing other MDSR system documents, such as maternal death review committee notes or meeting minutes from activities that occurred post-report dissemination, could provide a more complete picture of the status of MDSR systems and its use of data to inform action. Another aspect not explored in our study is how information from MDSR reports is shared with health providers, officers and others with influence to implement recommendations. Adding such information to the core content of national MDSR reports would be key to ensure engagement of the first-line cadre in implementing recommendations and strengthening the surveillance systems.

## Conclusion

Our review of MDSR country reports indicates that countries employ a variety of methods to measure key indicators such as notification and review rates in analyses of the MDSR program. Recommendations and subsequent response efforts are often not documented in follow-up reports. Comprehensive reporting on MDSR may help the system in accomplishing its core functions of improving accountability and quality of care. Reports need to include detailed descriptions of the system's key processes, particularly the response and recommendations that have been addressed, to what extent and if they are having the expected effect on maternal health. Ignoring suboptimal functioning of MDSR in national reports prevents countries in utilizing the system effectively and efficiently for reducing maternal deaths. There remain opportunities for developing a core reporting template, including guidance on how to analyze trends in key maternal mortality indicators, which would improve monitoring and evaluation of surveillance efforts and promote the use of MDSR data to prevent future maternal deaths.

## Supporting information

**S1 Text. Report content analysis items.**
(DOCX)

## Acknowledgments

The authors would like acknowledge the World Health Organization's MDSR Reporting and Monitoring Working Group for their input on this work: Charles Ameh (Charles. Ameh@lstmed.ac.uk); Luc de Bernis (lucdebernis1@gmail.com); Animesh Biswas (abiswas-s@unfpa.org); Louise Day (Louise-Tina.Day@lshtm.ac.uk); Bremen De Mucio (demuciob@-paho.org); Patricia Doherty (P.Doherty@options.co.uk); Pablo Duran (duranpa@paho.org); Temitayo Erogbogbo (temitayo.erogbogbo@msd.com); Debra Jackson (Debra.Jackson@lshtm.ac.uk); Neena Khadka (NKhadka@savechildren.org); Diane Morof (dmorof@cdc.gov) Sara Nam (Sara.GlobalHealth@gmail.com); Tedbabe Degefie Hailegebriel (thailegebriel@unicef.org); Endang Handzel (wuo5@cdc.gov); Fatima Gohar (fgohar@unicef.org); Minjoon Kim (mkim@unicef.org); Matthews Mathai (matthews.mathai@gmail.com); Allisyn Moran (morana@who.int), Moise Muzigaba (muzigabam@who.int); Francesca Palestra (palestraf@who.int); Maria Plotkyn (MPlotkin@fhi360.org); Kusum Thapa (Kusum.Thapa@jhpiego.org).

We would like to extend our deepest thanks to the UNFPA country offices that supplied most of the MDSR country reports reviewed for this study.

## Author Contributions

**Conceptualization:** Lillian Whiting-Collins, Florina Serbanescu, Ann-Beth Moller, Susanna Binzen, Jean-Pierre Monet, Michel Brun.

**Data curation:** Florina Serbanescu.

**Formal analysis:** Lillian Whiting-Collins, Florina Serbanescu, Susanna Binzen.

**Methodology:** Lillian Whiting-Collins, Florina Serbanescu, Ann-Beth Moller, Susanna Binzen, Jean-Pierre Monet, Michel Brun.

**Project administration:** Florina Serbanescu.

**Software:** Lillian Whiting-Collins.

**Supervision:** Florina Serbanescu, Michel Brun.

**Visualization:** Lillian Whiting-Collins, Susanna Binzen.

**Writing – original draft:** Lillian Whiting-Collins, Florina Serbanescu, Ann-Beth Moller, Susanna Binzen, Jean-Pierre Monet, Jenny A. Cresswell, Michel Brun.

**Writing – review & editing:** Lillian Whiting-Collins, Florina Serbanescu, Ann-Beth Moller, Susanna Binzen, Jean-Pierre Monet, Jenny A. Cresswell, Michel Brun.

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
