## [Decision Letter · Decision Letter 0]

24 May 2023

PGPH-D-23-00536

Maternal death surveillance and response system reports from 32 low-middle income countries, 2011–2020: what can we learn from the reports?

Dear Dr. Serbanescu,

Thank you for submitting your manuscript to PLOS Global Public Health. After careful consideration, we feel that it has merit but does not fully meet PLOS Global Public Health’s publication criteria as it currently stands. Therefore, we invite you to submit a revised version of the manuscript that addresses the points raised during the review process.

We look forward to receiving your revised manuscript.

Kind regards,

Anteneh Asefa Mekonnen, Ph.D., MPH

Academic Editor

Journal Requirements:

2. We noticed that you used "unpublished" in the manuscript. We do not allow these references, as the PLOS data access policy requires that all data be either published with the manuscript or made available in a publicly accessible database. Please amend the supplementary material to include the referenced data or remove the references.

3. In the online submission form, you indicated that "Data are available from the CDC Division of Reproductive Health upon reasonable request". All PLOS journals now require all data underlying the findings described in their manuscript to be freely available to other researchers, either 1. In a public repository, 2. Within the manuscript itself, or 3. Uploaded as supplementary information.

Additional Editor Comments (if provided):

Reviewers' comments:

Reviewer's Responses to Questions

**Comments to the Author**

1. Does this manuscript meet PLOS Global Public Health’s publication criteria? Is the manuscript technically sound, and do the data support the conclusions? The manuscript must describe methodologically and ethically rigorous research with conclusions that are appropriately drawn based on the data presented.

Reviewer #1: Yes

Reviewer #2: Yes

2. Has the statistical analysis been performed appropriately and rigorously?

Reviewer #1: Yes

Reviewer #2: Yes

3. Have the authors made all data underlying the findings in their manuscript fully available (please refer to the Data Availability Statement at the start of the manuscript PDF file)?

Reviewer #1: Yes

Reviewer #2: Yes

4. Is the manuscript presented in an intelligible fashion and written in standard English?

Reviewer #1: Yes

Reviewer #2: Yes

5. Review Comments to the Author

Reviewer #1: Maternal death surveillance and response system reports from 32 low-middle income countries, 2011–2020: what can we learn from the reports?

General comment:

This a generally well researched and written paper that was needed. The status of implementation of MDSR system across countries was largely unknown except from research that have been done in few countries. This study provides a glimpse on how countries have accepted and made efforts to implement the system.

Abstract

The abstract is well written

Background

It is a well written

Methods

1. Can the authors say what type of study was this? Was it qualitative study or a mix quantitative-qualitative study?

2. Its not clear why the author decided to include the reports before 2011, while the MDSR system was initiated by the WHO in 2013

3. It is reported that some countries were excluded since they did not meet the inclusion criteria, is it possible to mention the countries that were excluded? Do ethics allow this? I think it would be good for these countries to know why they were excluded so that they can improve on their reporting

4. The author explained well what was assessed and the data extraction guide which was obtained from the WHO guidance on dissemination and reporting. Its not clear whether the WHO guidance provides guide for reporting each item. Are all the countries using this guide or does each country decide how to report each item?

5. The authors should also explain how the recommendations were assessed. Was the SMART criteria or any other criteria used??

Results

1. Line 215: Some countries reported to have “complete” data. How did these countries assess the completeness. It not clear in general how the countries assessed completeness of their data. Is there guide in the WHO guidance on how to do this?

2. Figure 1 is not well visible

3. The text in all other figures in blurred. The author can provide figures with better resolution

Discussions

1. It’s a well written discussion

2. It would benefit with more references to show what research on MDSR in different countries have reported

3. The authors should also discuss on reporting on how the reports were created and stakeholders involved in making the reports. This is another crucial element in making sure the reports reflect the situation in the countries with no biases or political influence in the reports.

Reviewer #2: The authors reviewed MDSR reports from low-and-middle income countries (LMICs) to examine core content and

identify how surveillance data and data dissemination could be improved to guide recommendations and track responses. A deductive content analysis of 56 MDSR reports from 32 LMICs was conducted. Among other findings, the authors found that slightly over half of the reports described their data as incomplete and reported deaths at both community and facility levels, while a third calculated review completion rates. Differences were observed in how notification and review rates were calculated.

This is a very important evidence contributing to strengthening MDSR.

The authors have described the methods used comprehensively and have presented the results appropriately.

It would be good to also show the observed differences by region for further exploration and future understanding in case of major differences in the observations.

Otherwise, I have no further comments.

6. PLOS authors have the option to publish the peer review history of their article (what does this mean?). If published, this will include your full peer review and any attached files.

**Do you want your identity to be public for this peer review?** For information about this choice, including consent withdrawal, please see our Privacy Policy.

Reviewer #1: No

Reviewer #2: **Yes: **Dr Choolwe Jacobs

---

## [Decision Letter · Decision Letter 1]

13 Oct 2023

PGPH-D-23-00536R1

Maternal death surveillance and response system reports from 32 low-middle income countries, 2011–2020: what can we learn from the reports?

Dear Dr. Serbanescu,

Thank you for submitting your manuscript to PLOS Global Public Health. After careful consideration, we feel that it has merit but does not fully meet PLOS Global Public Health’s publication criteria as it currently stands. Therefore, we invite you to submit a revised version of the manuscript that addresses the points raised during the review process.

Your manuscript has been evaluated by one new reviewer; their comments are appended below and included in the attached file.

The reviewer has commented on matters throughout your manuscript, particularly in relation to the true functioning of the MDSR system and how the reality of the MDSR system functionality may affect your findings and conclusions as they pertain to improving maternal health. The reviewer has also provided other comments, as well as advice regarding the use of some terminology and phrasing; please ensure you address each of the reviewer's comments when revising your manuscript.

We look forward to receiving your revised manuscript.

Kind regards,

Hugh Cowley

Staff Editor

Journal Requirements:

Additional Editor Comments (if provided):

Reviewers' comments:

Reviewer's Responses to Questions

**Comments to the Author**

1. If the authors have adequately addressed your comments raised in a previous round of review and you feel that this manuscript is now acceptable for publication, you may indicate that here to bypass the “Comments to the Author” section, enter your conflict of interest statement in the “Confidential to Editor” section, and submit your "Accept" recommendation.

Reviewer #3: (No Response)

2. Does this manuscript meet PLOS Global Public Health’s publication criteria? Is the manuscript technically sound, and do the data support the conclusions? The manuscript must describe methodologically and ethically rigorous research with conclusions that are appropriately drawn based on the data presented.

Reviewer #3: Partly

3. Has the statistical analysis been performed appropriately and rigorously?

Reviewer #3: N/A

4. Have the authors made all data underlying the findings in their manuscript fully available (please refer to the Data Availability Statement at the start of the manuscript PDF file)?

Reviewer #3: Yes

5. Is the manuscript presented in an intelligible fashion and written in standard English?

Reviewer #3: Yes

6. Review Comments to the Author

Reviewer #3: The manuscript obviously addresses an important public health issue in maternity care. My major problem is that the conclusion of the abstract is far too vague and is not giving the only conclusion which can be made of this review. My comment is repeated at the conclusion of main manuscript: the most important conclusion from this study must be the suboptimal functioning of the MDSR system, which therefore at present will have no or very limited impact of improving maternal health. In the background of the abstract, you wrote: Maternal Death Surveillance and Response (MDSR) systems generate information that aid efforts to end preventable maternal deaths". Unfortunately we have no clear evidence of this too bold statement, I therefore changed this in "may aid". It is not just all the paperwork written in this paper in a somewhat bureaucratic way, that will have any influence on maternal health. But I would agree that when the whole health system functions in all possible ways, MDSR may aid to improve maternal health. But this review by no way could convince anybody with experience in the field (I mean with his feet in the clay) because of the suboptimal functioning of MDSR could do anything, while and I agree in principle it could help, when all other parts of the system are in place and function well. This has to made clear in the manuscript

There are many more minor issues which can be found in the uploaded and edited manuscript. In my view it is better to give everywhere the numbers of the reports with % between brackets. One last issue: women give birth and pizzas are delivered and services exist in shops; in health facilities care is given

7. PLOS authors have the option to publish the peer review history of their article (what does this mean?). If published, this will include your full peer review and any attached files.

**Do you want your identity to be public for this peer review?** For information about this choice, including consent withdrawal, please see our Privacy Policy.

Reviewer #3: **Yes: **jos van roosmalen

---

## [Decision Letter · Decision Letter 2]

1 Dec 2023

PGPH-D-23-00536R2

Maternal death surveillance and response system reports from 32 low-middle income countries, 2011–2020: what can we learn from the reports?

Dear Dr. Serbanescu,

Thank you for submitting your manuscript to PLOS Global Public Health. After careful consideration, we feel that it has merit but does not fully meet PLOS Global Public Health’s publication criteria as it currently stands. Therefore, we invite you to submit a revised version of the manuscript that addresses the points raised during the review process.

The manuscript has been evaluated by one of the previous reviewers, and their comments are available below. The reviewer has some minor outstanding concerns that are indicated in the attached, annotated version of your manuscript. Could you please revise the manuscript to address the concerns raised?

We look forward to receiving your revised manuscript.

Kind regards,

Paraminder Dhillon, PhD

Staff Editor

Journal Requirements:

2. Please amend your Data Availability Statement and indicate where the data may be found.

Additional Editor Comments (if provided):

Reviewers' comments:

Reviewer's Responses to Questions

**Comments to the Author**

1. If the authors have adequately addressed your comments raised in a previous round of review and you feel that this manuscript is now acceptable for publication, you may indicate that here to bypass the “Comments to the Author” section, enter your conflict of interest statement in the “Confidential to Editor” section, and submit your "Accept" recommendation.

Reviewer #3: All comments have been addressed

2. Does this manuscript meet PLOS Global Public Health’s publication criteria? Is the manuscript technically sound, and do the data support the conclusions? The manuscript must describe methodologically and ethically rigorous research with conclusions that are appropriately drawn based on the data presented.

Reviewer #3: Yes

3. Has the statistical analysis been performed appropriately and rigorously?

Reviewer #3: N/A

4. Have the authors made all data underlying the findings in their manuscript fully available (please refer to the Data Availability Statement at the start of the manuscript PDF file)?

Reviewer #3: Yes

5. Is the manuscript presented in an intelligible fashion and written in standard English?

Reviewer #3: Yes

6. Review Comments to the Author

Reviewer #3: almost all comments have been adequately adressed, apart from a few, which cab be found in the attached file

7. PLOS authors have the option to publish the peer review history of their article (what does this mean?). If published, this will include your full peer review and any attached files.

**Do you want your identity to be public for this peer review?** For information about this choice, including consent withdrawal, please see our Privacy Policy.

Reviewer #3: **Yes: **jos van roosmalen

---

## [Editor Report · Decision Letter 3]

27 Dec 2023

Maternal death surveillance and response system reports from 32 low-middle income countries, 2011–2020: what can we learn from the reports?

PGPH-D-23-00536R3

Dear Dr Serbanescu,

We are pleased to inform you that your manuscript 'Maternal death surveillance and response system reports from 32 low-middle income countries, 2011–2020: what can we learn from the reports?' has been provisionally accepted for publication in PLOS Global Public Health.

Best regards,

Julia Robinson

Executive Editor